# Enhancing Coleoptile Length of Rice Seeds under Submergence through *NAL11* Knockout

**DOI:** 10.3390/plants13182593

**Published:** 2024-09-17

**Authors:** Zhe Zhao, Yuelan Xie, Mengqing Tian, Jinzhao Liu, Chun Chen, Jiyong Zhou, Tao Guo, Wuming Xiao

**Affiliations:** 1National Engineering Research Center of Plant Space Breeding, South China Agricultural University, Guangzhou 510642, China; zhezhao3016@163.com (Z.Z.); 18110520310@163.com (M.T.); ljz666666@stu.scau.edu.cn (J.L.); chchun@scau.edu.cn (C.C.); 2Yangjiang Institute of Agricultural Sciences, Yangjiang 529500, China; m13751844286@163.com; 3Guangdong Agricultural Technology Extension Center, Guangzhou 510520, China; nytzjy@126.com

**Keywords:** rice seed germination, coleoptile elongation, submergence, *NAL11*, stress tolerance

## Abstract

Submergence stress challenges direct seeding in rice cultivation. In this study, we identified a heat shock protein, *NAL11*, with a DnaJ domain, which can regulate the length of rice coleoptiles under flooded conditions. Through bioinformatics analyses, we identified *cis*-regulatory elements in its promoter, making it responsive to abiotic stresses, such as hypoxia or anoxia. Expression of *NAL11* was higher in the basal regions of shoots and coleoptiles during flooding. *NAL11* knockout triggered the rapid accumulation of abscisic acid (ABA) and reduction of Gibberellin (GA), stimulating rice coleoptile elongation and contributes to flooding stress management. In addition, *NAL11* mutants were found to be more sensitive to ABA treatments. Such knockout lines exhibited enhanced cell elongation for coleoptile extension. Quantitative RT-PCR analysis revealed that *NAL11* mediated the gluconeogenic pathway, essential for the energy needed in cell expansion. Furthermore, *NAL11* mutants reduced the accumulation of reactive oxygen species (ROS) and malondialdehyde under submerged stress, attributed to an improved antioxidant enzyme system compared to the wild-type. In conclusion, our findings underscore the pivotal role of *NAL11* knockout in enhancing the tolerance of rice to submergence stress by elucidating its mechanisms. This insight offers a new strategy for improving resilience against flooding in rice cultivation.

## 1. Introduction

Rice (*Oryza sativa* L.) stands as one of the most pivotal crops cultivated globally, with over half of the world’s population relying on it as a primary staple food [1]. The practice of direct seeding plays a crucial role in rice cultivation and is widely employed in both rainfed and irrigated fields due to its substantial benefits, including reductions in labor, energy consumption, water usage, production costs, and mechanization [2,3]. However, rice’s vulnerability to prolonged flooding poses a significant challenge, leading to oxygen starvation and energy depletion in submerged plants [4]. Enhanced seedling vigor, characterized by the elongation of mesocotyls, coleoptiles, and shoots, is crucial for improving seedling emergence under such conditions [5,6]. Rice employs an escape strategy to reduce submergence stress during seed germination [7], increasing coleoptile and/or mesocotyl length to improve survival under submerged conditions [8,9].

Several genetic factors affect rice shoot growth, such as expansin genes, anaerobic metabolic pathways including glycolysis and fermentation, ROS scavenging and phytohormone signaling. Overexpression of the expansin gene *OsEXP4* has been shown to promote mesocotyl and coleoptile elongation by cell wall stress relaxation and volumetric extension, a process that is repressed in *OsEXP4-antisense* plants [10]. In addition, up-regulation of *EXPA7* and *EXPB12* promotes the elongation of rice coleoptile under hypoxic conditions [11]. In rice, sugar availability has been considered one of the critical factors for tolerance to submergence [12]. Rice seeds can germinate and produce α-amylase enzymes required for starch degradation even without oxygen [13]. Under flooded conditions, anaerobic metabolic pathways, including glycolysis and fermentation, play a crucial role in coleoptile elongation [14]. Under sugar starvation, the transcription factor *MYBS1* activates the *Ramy3D* gene, facilitating starch degradation to provide the necessary energy for subsequent leaf and root development [15]. During germination under submergence, rice gene *CIPK15* (calcineurin B-like-interacting protein kinase 15) regulates coleoptile length through a sugar signaling pathway [16]. Alcohol dehydrogenase (ADH) activity in rice coleoptiles is correlated with a deceleration in coleoptile elongation under submergence conditions [17].

Abiotic stresses, like drought, salt, and temperature variations, enhance the production of ROS in plants [18]. However, excessive ROS can lead to oxidative damage to lipids, DNA, and proteins [19]. To mitigate ROS-induced damage, plants have developed an antioxidant system consisting of enzymes, such as catalase (CAT), ascorbate peroxidase (APX), glutathione peroxidase (GPX), glutathione reductase (GR), and glutathione sulfotransferase (GST) [20]. Abscisic acid (ABA) is a pivotal stress hormone that accumulates in response to stress and concurrently associated with a reduction in growth in stressed plants [21]. However, a growing body of evidence suggests that ABA plays a dual role in plant stress responses; while high concentrations inhibit growth, low concentrations can promote it [22,23]. This balance is crucial in stressed plants, where ABA concentrations are finely tuned through a balance between ABA biosynthesis and catabolism processes [24]. Recent studies have shown that the crosstalk between ABA and ROS in the phytohormone network play important roles in many aspects of plant growth and development, including the response to adversity stresses [25,26]. For example, it has been reported that alterations in ROS levels can affect ABA biosynthesis and signaling, as well as change ABA sensitivity [27], and ABA can also regulate the expression of ROS producing and scavenging genes [25]. For instance, overexpression of the rice ABA receptor 6 (*OsPYL6*) can improve drought tolerance by increasing ABA content and improving ROS detoxification, thereby stabilizing membrane [28]. Studies had suggested that ABA could improve oxidase activity and induce stomatal closure to reduce CO2 fixation, thereby inhibit the accumulation of ROS [29]. In addition to ABA, other phytohormones play crucial roles in regulating growth and developmental process and signaling networks involved in plant responses to environmental stresses, including flooding [30]. Gibberellin (GA) is considered essential in regulating the expression of α-amylase genes, which catalyse hydrolytic starch degradation during cereal seed germination in the air. However, under anoxic conditions, starch degradation through the gibberellin-induced α-amylase pathway fails to function properly because oxygen is also required for gibberellin biosynthesis, and rice become gibberellin insensitive under anoxic or hypoxic conditions [12]. Auxin is well known for promoting coleoptile elongation and rapid seedling growth during germination [31], but little is known about its role in rice germination and seedling establishment under submergence. A recent study has shown that auxin biosynthesis and the auxin influx carrier AUX1 regulated the final length of rice coleoptile under submergence [32].

Heat-shock proteins are proteins with molecular chaperone activity, responsible for protein folding, assembly, translocation and degradation in many normal cellular processes, stabilize proteins and membranes [33]. And HSPs may be newly synthesized or otherwise increase in abundance in vivo when plants are subjected to stress. Heat-shock proteins have been reported to play critical roles in stress resistance. For example, transgenic rice plants overexpressing *sHSP17.7* showed increased survival under high-temperature conditions [34]. The overexpression of *Hsp70* genes positively correlates with the acquisition of thermotolerance [27] and results in enhanced tolerance to salt, water and high-temperature stress in plants. *HSP70s* reduce heat tolerance, but under high-salt conditions, these proteins enhance seed germination and regulate the developmental transition from seed to seedling by repressing seed-specific gene expression [35]. The expression of *Hsp90* in Arabidopsis is developmentally regulated and responds to heat, cold, salt stress, heavy metals, phytohormones, and light/dark transitions [36]. A correlation between HSPs and anaerobiosis has been observed in the hearts of turtles and mammals, where constitutive expression of certain HSP genes is associated with increased tolerance to anoxia [37].

Coleoptile and mesocotyl elongation are critical for rice survival under submerged conditions, and these processes are closely linked to hormonal regulation (ABA, GA, auxin) [3,32]. Our previous studies demonstrated that *NAL11*, which encodes a heat shock protein containing the DnaJ structural domain, regulates rice plant architecture and is involved in GA metabolism [38], showing the influence of *NAL11* on hormone regulation. This prompted us to investigate whether the regulatory function of *NAL11* in growth would intersect with stress response pathways, particularly under environmental stresses such as flooding, which can challenge plant survival. Therefore, this study aims to investigate the effect of *NAL11* on the elongation of rice coleoptile under flooded conditions. To further explore the interactions between *NAL11* and gibberellins, auxins and abscisic acid under submergence stress. Our findings unveil a novel mechanism by which HSPs contribute to flood tolerance in rice, which lays a foundation for further investigations into flooding tolerance in rice.

## 2. Results

### 2.1. Bioinformatics Characteristics

All amino acid and nucleotide sequences were retrieved and downloaded from the NCBI website. Using the NCBI blast online comparison, 28 homologous proteins were screened in *Oryza sativa Japonica*, *Brachypodium distachyon*, *Sorghun bicolor*, *Zea mays*, *Glycine max*, *Nymphaea colorata*, *Arabidopsis*, and wild rice (*Oryza brachyantha*, *Oryza glaberrima*). To comprehensively characterize *NAL11* and its homologous proteins, a phylogenetic tree was constructed for the 28 homologous proteins and the NAL11 protein using MEGA software (v11.0). The constructed evolutionary tree was then merged with the phylogenetic tree of the conserved structural domain elements of the proteins using the TBtools software (v1.120). Remarkably, in our phylogenetic analysis showed that NAL11 (XP_015645205.1) and XP_040381942.1 clustered together on the same branch (Figure 1). NP_001412649.1 was found to be orthologous to *NAL11* in rice, suggesting that *NAL11* has undergone a gene duplication event during evolution. Using the online software MEME (v5.5.7) was used to analyze the 28 homologous proteins, Motif 1 was identified in all 28 members, with conserved motifs 1, 2, and 3 in 22 encoding proteins (Figure 1). All homologues of *NAL11*, except for NP_001146964.1, contained the DnaJ structural domain (Appendix A). In addition to its conserved nature, NAL11 exhibited high homology with homologues in wild rice. These findings highlight *NAL11* as a highly conserved gene throughout evolution. To further investigate potential regulatory mechanisms, we analyzed the 2.0-kb nucleotide sequences upstream of the start codon using Plant CARE. This revealed multiple cis-elements associated with stress response (Appendix A), including the ABRE-motif (ABA response element), the TGACG-motif (MeJA-responsiveness), the GGTCCAT-motif (Auxin-responsiveness), the TCTGTTG-motif (Gibberellin-responsiveness), and other elements critical for endosperm expression, anaerobic induction, anoxic specific regulation, and meristem expression. These results suggest that the *NAL11* gene may respond to hypoxia or anoxia during rice growth and development.

### 2.2. The Expression of NAL11 Was Induced during Seed Germination under Submerged Conditions

Under aerobic conditions, *NAL11* exhibited a consistently low level of expression, with a discernible decline at both 12 h and 72 h compared to the baseline at 0 h (Figure 2A). In contrast, during submergence, the gene was induced to express at high levels at 12 h, 24 h, 48 h, and 60 h, compared to the corresponding expression levels under submerged conditions. This finding highlights a substantial upregulation in the expression of *NAL11* under submergence conditions compared to that under aerobic conditions. Peak expression occurred at 60 h under submergence, representing an approximately 3-fold increase compared to the expression levels under aerobic conditions. Despite a subsequent decrease in the expression at 72 h under submergence, the level remained significantly elevated compared to the expression level observed under aerobic conditions.

The expression pattern was assessed in transgenic plants carrying the *pNAL11::GUS* construct (GUS reporter gene driven by the promoter of the *NAL11* gene) in the ZH11 background under both submerged and aerobic conditions (Figure 2C). The analysis revealed that *NAL11* was predominantly expressed in the parenchyma tissues of the protruding embryo at 24 h and 48 h under aerobic conditions. Whereas at 72 h after seed germination, however, its expression was mainly observed in the growing bud, with minimal presence in the developing radicle. Conversely, under submergence conditions, *NAL11* exhibited distinct expression patterns. At 24 h, expression was concentrated in the protruding embryo, and extended to both the embryo and the growing bud at 48 h. Under submerged conditions, the expression in the parenchyma tissues of the embryo was prominent. Due to the limited observation of radicles during seed germination under submergence, *NAL11* expression was mainly observed in the elongating coleoptile, with a more pronounced signal closer to the base.

### 2.3. Knockout Lines Exhibited Longer Coleoptiles during Seed Germination under Submerged Conditions

We used two independent homozygous hygromycin-free transgenic lines with different editing effects (*osnal11-1* with 1 bp deletion and *osnal11-2* with 1 bp insertion) on the target gene (Appendix A). The *NAL11* knockout lines had significantly lower transcript levels than the wild type (Appendix A). Mature dry seeds of the WT (ZH11) and two knockout lines were subjected to germination under both air (aerobic) and waterlogged (anoxic) conditions. The coleoptile length of dry seeds on day 4 under submergence exhibited significant differences between the WT and the two knockout lines (Appendix A). The average coleoptile length of the two knockout lines reached 4.64 cm and 4.55 cm, respectively, which was significantly greater than the 3.82 cm observed in the WT (Appendix A). Moreover, the surface area of the coleoptile in the knockout lines was significantly greater than that observed in the WT (Appendix A). However, no evident differences in coleoptile diameter were observed between the WT and knockout lines (Appendix A). Remarkably, by day 4, both the WT and knockout lines achieved an impressive germination rate of approximately 100% on day 4 (Appendix A). Additionally, when germinated under aerobic conditions on day 4, no evident differences in bud and root lengths were observed between the WT and knockout lines (Appendix A). In conclusion, knockout of *NAL11* significantly improved rice seed coleoptile growth under submerged conditions.

It has previously demonstrated that coleoptile elongation under anoxic conditions is attributed to cell expansion, and that expansins are likely to be key players in this physiological process [10,39]. To further understand the role of expansins in this context, we compared the expression patterns of some expansin genes in ZH11 and knockout lines after 48 h of flooding. Interestingly, only *OsEXP7* showed downregulation in the coleoptiles of the knockout lines at 48 h. In contrast, the mRNA levels of *OsEXP2*, *OsEXP4*, *OsEXP8*, *OsEXPB11*, *OsEXPB12*, and *OsEXPB15* were significantly higher in the coleoptiles of the knockout lines after submerged germination compared to those of the WT at the same time point (Figure 3), with *OsEXP4* showing an impressive approximately 4-fold increase (Figure 3). The notable upregulation of the expression of these expansin genes in the knockout lines may provide a molecular basis for the accelerated growth of rice coleoptiles under submerged conditions compared to that in the WT. This suggests that expansins are likely to contribute to the elongation of coleoptiles in rice when exposed to anoxic conditions.

### 2.4. Knockout of NAL11 Affected Sugar and Energy Pathways under Submerged Conditions

As a crucial enzyme responsible for catalyzing the degradation of starch in cereal seeds, α-AMS plays a pivotal role in various key agronomic traits, including the germination rate [40] and resistance to hypoxia stress [41]. In our study, we aimed to identify the specific α-AMS family member directly involved in the response to endosperm starch degradation under submergence stress. To achieve this, we investigated the expression of rice *α-AMS* genes in seeds (containing coleoptiles) of ZH11 and knockout lines at 48 h after submergence treatment (Figure 4A). Our findings revealed that the expression levels of *RAmy1A* and *RAmy2A* in the knockout lines were significantly lower than those in ZH11. Conversely, the transcript levels of *Ramy3A*, *RamyC*, and *RAmy3D* were significantly higher in the knockout lines than ZH11 and that of *RAmy3A* in the knockout lines was approximately 5-fold higher in the knockout lines than in ZH11. Interestingly, no evident differences were observed in the expression levels of *RAmy3C* and *RAmy3E* between ZH11 and the knockout lines. Furthermore, the α-AMS activity in both knockout lines and WT showed a consistently significant increase from 12 h to 96 h during submerged germination. Compared to the WT, the knockout lines consistently showed higher α-AMS activity at different stages (Figure 4B).

Long-term submergence can lead to substantial carbohydrate consumption, resulting in energy deficiency [42]. Therefore, we investigated the expression patterns of glycolytic pathway genes in rice seeds (containing coleoptiles) under submerged conditions (Figure 4C). *CDPK15*, *MYBS1*, *Adh1*, and *PDC1* exhibited distinct submergence-dependent expression. The expression of *CDPK15* decreased in the knockout lines, while other key genes associated with energy pathways, including *MYBS1*, *Adh1*, and *PDC1*, showed a significant upregulation in the knockout lines at 48 h after submergence treatment. Concurrently, the enzyme activity of α-AMS exhibited a similar increasing trend mirroring the transcript levels of the energy-synthesizing genes. The observed expression patterns of these genes suggest that the knockout of *NAL11* may have a substantial impact on the regulation of *α-AMS* genes and energy-synthesizing genes, potentially affecting the plant’s ability to cope with submergence stress.

### 2.5. Knockout of NAL11 Affects ROS Levels and the Expression of Some Stress-Related Genes under Submerged Conditions

Given the anaerobic stress experienced by both the knockout lines and the WT during submergence, it is imperative to investigate into the potential physiological changes at different stages, aiming to elucidate he differences in coleoptile length. ROS are important signals that regulate the stress tolerance. Here, we compare the accumulation of hydrogen peroxide (H_2_O_2_) and MDA between seeds of the WT and knockout lines after submergence treatment. It shows that the seeds of the knockout lines accumulated more MDA and H_2_O_2_ than the WT (Figure 5A,B). Considering that *OsRbohA* and *OsRbohE* belong to the NOX family [43], which is key to the production of ROS. We determined the transcription level of these two genes. We found that both were reduced in the knockout lines compared to WT (Figure 5C,D), which was consistent with ROS levels.

In addition, we also determined the activities of ROS-scavenging enzymes (SOD, POD, CAT). The results showed that there was no significant difference between WT and knockout lines in the control group. CAT activity in the knockout lines was evidently lower than that in the WT at the beginning (0 h). Surprisingly, the knockout lines exhibited significantly higher CAT activity than the WT at 12 h, 72 h, and 96 h of seed germination under submergence, with no apparent differences at 24 h and 48 h (Appendix A). Moreover, the knockout lines showed significantly increased SOD activity from 0 h to 96 h under submerged conditions compared to the WT (Appendix A). However, no evident difference was observed at 48 h. In contrast, POD activity was increased in both knockout lines and WT after submergence compared to that at the baseline (0 h). From 12 h to 96 h under submergence, no evident differences in POD activity were observed between the knockout lines and WT (Appendix A). The above results indicated that knockout of *NAL11* can enhance rice tolerance to submergence stress by improving ROS scavenging ability.

To further investigate the possible molecular mechanisms of *NAL11* in regulating submergence tolerance in plants, we also determined the transcript levels of some well-known stress-responsive genes. These included *OsSUB1A* and *OsNAC9*, encoding typical stress-related NAC-type transcription factors (TFs); *OsSnRK1A* and *OsTPP7*, encoding trehalose-6-phosphate(T6P) phosphatasegene (*OsTPP7*) proteins. After submergence treatment for 48 h, compared to WT, the mRNA levels of the above genes were both increased in knockout lines (Appendix A).

### 2.6. NAL11 Is Involved in the Phytohormone-Mediated Regulatory Pathway

Previous studies have revealed that plant hormones are signaling compounds that regulate crucial aspects of growth, development and environmental stress responses [43]. The contents of several endogenous phytohormones, including ABA, IAA and GA, were measured in the WT and knockout lines to determine the regulatory pathway in which *NAL11* is involved. There were no significant differences in endogenous ABA, GA and IAA concentrations between the WT and knockout lines at 24 h after the initiation of submergence treatment. However, as the submergence time increased, a significant increase in ABA and IAA concentrations was observed in the knockout lines, whereas no significant change was observed in the WT (Figure 6A). The knockout lines showed a significantly higher ABA and IAA levels than the WT at 48 h and 72 h after submergence treatment, while the opposite was true for GA (Figure 6B).

Therefore, we further analyzed the expression of genes related to ABA, GA and IAA synthesis and metabolism in the WT and knockout lines at 24 h, 48 h and 72 h after submergence. Consistent with the increase in the ABA and IAA levels, most of the genes related to ABA and IAA synthesis in the knockout lines were upregulated (Appendix A). Moreover, most of the genes related to ABA metabolism, including several genes of the *ABAox* family genes, were downregulated simultaneously (Appendix A). To further elucidate the effects of *NAL11* knockout on the ABA signaling pathway, we examined the expression levels of key genes involved in ABA regulation and signaling. Since *OsDET1*, *OsbZIP46*, and *OsbZIP72* have been identified as positive regulators of ABA signaling [44], our study specifically focused on analyzing the expression levels of *OsDET1*, two *bZIP* genes (*OsbZIP46* and *OsbZIP72*) involved in ABA regulation, and several ABA receptor genes in both the WT and knockout lines under anaerobic conditions. At 48 h after submergence, the knockout lines exhibited a significant increase in the expression of *OsbZIP72*, whereas no significant differences were observed in the expression levels of *OsDET1* and *OsbZIP46* between the WT and knockout lines (Appendix A). Moreover, *OsPYL1*, *OsPYL2*, *OsPYL3*, *OsPYL8*, and *OsPYL10* were significantly upregulated compared to the WT, whereas *OsPYL5* and *OsPYL9* were downregulated in the knockout lines. In contrast, no significant changes were observed in the transcript levels of *OsPYL4* and *OsPYL7* in the WT and knockout lines (Appendix A). At 72 h after submergence, most of the genes related to ABA signaling were upregulated (Appendix A). This suggests that knockout of *NAL11* has a significant effect on the expression of genes related to ABA biosynthesis, catabolism and signal transduction processes. Interestingly, the expression of most GA-related genes in the *NAL11* knockout lines showed the opposite trend compared to ABA (Appendix A), indicating reduced GA activity. We therefore speculate that the longer coleoptiles of the *NAL11* knockout lines under submergence could be attributed to the upregulated expression of ABA- and IAA-related genes and down-regulated expression of GA-related genes. Furthermore, we also analyzed the expression of genes related to auxin biosynthesis in one of the knockout lines. The expression of *YUCCA2*, *YUCCA3* and *YUCCA6*, which are involved in auxin biosynthesis, was significantly increased in the knockout line compared to the WT under submerged conditions (Appendix A). Besides, the expression pattern of *TAA1*, another phytoalexin biosynthesis gene, was also similar to that of these genes (Appendix A). As a result, the endogenous IAA levels were significantly increased (Figure 6C). It suggests that under submerged conditions, *NAL11* knockout may prolong the coleoptile by increasing the accumulation of auxin in rice seeds. Taken together, *NAL11* is involved in the ABA, GA and auxin pathways to improve tolerance to submergence stress tolerance in rice. This intricate regulatory network contributes to the improved tolerance of the knockout lines to submergence stress during the germination stage of rice seeds.

### 2.7. Knockout Lines Are Sensitive to ABA

Crosstalk between ABA and ROS in the phytohormone network has been shown to be involved in the regulation of plant stress tolerance [29], and exogenous ABA treatment will decrease the level of ROS in rice seed germination embryos [45]. Based on these studies, we conducted the following experiment to assess the sensitivity of *NAL11* to ABA. We conducted a statistical analysis of the growth status of the WT and knockout lines on day 4 after treatment with different concentrations of exogenous ABA (0 µM, 0.001 µM, 0.01 µM, 0.1 µM, 1 µM, and 10 µM). Under submerged conditions, the knockout lines exhibited longer coleoptiles compared to the WT in the absence of ABA treatment. However, the coleoptile length of the knockout lines was significantly reduced compared to that of the WT when treated with higher concentrations (10 µM, 1 µM, or 0.1 µM) of exogenous ABA (Figure 7A,B). Remarkably, at an ABA concentration of 0.01 µM, no significant difference in coleoptile length was observed between the WT and knockout lines. Interestingly, when treated with a lower concentration of 0.001 µM ABA, the coleoptile length of the knockout lines was significantly greater than that of the WT. These results suggest that high concentrations of exogenous ABA inhibit coleoptile elongation in the knockout lines, while low concentrations of exogenous ABA promote coleoptile elongation in the knockout lines. This suggests that the knockout of *NAL11* may enhance sensitivity to ABA.

## 3. Discussion

The length and elongation rate of the coleoptile and/or mesocotyl are the crucial developmental traits that determines deterring the success of direct seeding in many cereal crops. In the case of rice submerged seed germination, the long and rapidly elongating coleoptiles may promote submergence tolerance by providing oxygen when they reach to the water surface [10]. Interestingly, the high expression of some HSPs in response to hypoxia is also a prominent phenomenon. This observation suggests a potential close relationship between *NAL11* and hypoxia stress responses. Supporting this hypothesis, our study suggests the involvement of at least four biological processes in *NAL11*-regulation of hypoxia response, encompassing substance metabolism, abiotic stress responses, redox reactions, and crosstalk with other hormones. While this highlights the involvement of *NAL11* in hypoxia response, we acknowledge that it may not serve as a key integrator of these pathways. Rather, *NAL11* may contribute to stress responses through its regulatory role within these processes. By identifying hypoxia-responsive genes and pathways, our research contributes to the understanding of the escape strategy of submergence tolerance in rice, which may be helpful in improving submergence tolerance in other cereal crops.

Expansin genes, known for their role in promoting cell wall relaxation and expansion, were significantly upregulated in the *NAL11* knockout lines. Previous research has demonstrated that overexpression of expansin genes, such as *OsEXP4*, enhances coleoptile and mesocotyl elongation in rice [10]. Consistent with these findings, we observed significant upregulation of expansin genes in the knockout lines, which suggests that *NAL11* may negatively regulate coleoptile elongation by controlling expansin activity. This regulatory effect on cell elongation is likely a key mechanism by which *NAL11* influences submergence tolerance during the germination stage. Additionally, our analysis revealed the presence of various response-related cis-elements in the promoter of the *NAL11* gene (Appendix A), including ABA, GA, and auxin response elements, further supporting its involvement in hormonal regulation (Figure 6 and Appendix A). In this study, we were the first to investigate the role of *NAL11*, a DnaJ domain-containing HSP gene, in regulating coleoptile elongation through expansin expression and hormone crosstalk. Reactive oxygen species (ROS) function as crucial signaling molecules, but excessive ROS accumulation can cause irreversible cell damage [46]. Plants adapt to abiotic stress by regulating ROS metabolism [47,48]., and our results showed enhanced activities of antioxidant enzymes (CAT and SOD) in *NAL11* knockout lines under submergence stress, although POD activity was not significantly affected (Appendix A). This increase in antioxidant enzyme activity likely contributes to the enhanced submergence tolerance observed in the knockout lines [49,50,51]. Furthermore, the knockout lines exhibited reduced levels of MDA, a marker for ROS-induced lipid peroxidation [52], indicating that *NAL11* knockout may enhance antioxidant capacity to mitigate oxidative stress.

Energy supply is crucial for rice seed germination under anoxic conditions, where starch is degraded into fermentable carbohydrates to sustain embryo growth [53]. Our results showed that amylase activity was significantly higher in NAL11 knockout lines during submergence, which is consistent with longer coleoptiles observed in these lines (Appendix A). This increased amylase activity likely facilitates more efficient starch hydrolysis, contributing to submergence tolerance. Notably, expression levels of key amylase genes, including RAmy3A, RAmyC, and RAmy3D, were upregulated in the knockout lines under submergence, suggesting an accelerated starch metabolism (Figure 4B). These findings align with previous studies showing that the upregulation of amylase genes supports seed germination and early seedling establishment under submerged conditions [42,54]. Although the expression of *RAmy1A* and *RAmy2A* was reduced after submergence (Figure 4A), this did not affect the rate of band starch hydrolysis, as the isozymes they encode are not dominant during hypoxic sprouting [11,55]. α-AMS 3 emerges as a major player, its mRNA accounting for approximately 60% of the total mRNA of amylase genes in glucose-starved rice cells [56]. In this study, high expression of α-AMS 3 accelerated the hydrolysis of starch (Figure 4A), thereby providing the energy required for the germination process and sustaining the subsequent alcoholic fermentation process. Previous studies have shown that submergence triggers sugar starvation and induces mRNA accumulation of calcineurin B-like (CBL) protein-interacting protein kinase 15 (*CIPK15*), thereby enhancing the accumulation of SnRK1A proteins. These two proteins interact and induce the *MYBS1* transcription factor, subsequently activating the expression of starvation-induced α-amylase gene, *αAmy3/RAmy3D* [16]. As expected, the expression of *CIPK15* and *MYBS1* was significantly higher in the knockout lines compared to WT (Figure 4C). Alcoholic fermentation plays a crucial role in providing ATP under hypoxic conditions [57,58], and our results showed that *NAL11* knockout plants exhibited higher expression of pyruvate decarbox-ylase (PDC) and ADH after 48 h of submergence (Figure 4C), which supports glycolysis and ATP synthesis [59]. This observation suggests that *NAL11* knockout enhances rice’s ability to cope with submergence through more efficient anaerobic metabolism. But it remains unclear whether sugar could affect the accumulation of endogenous free phytohormones to influence submergence tolerance and seedling establishment in rice.

Many transcription factors play critical roles in regulating the stress response in plants, including *SUB1A* and *OsNAC9*, which improve submergence tolerance when overexpressed [60,61]. Under sugar starvation conditions, SnRK1A is an important mediator in the sugar signaling cascade response, acts upstream of *MYBS1* and *αAmy3* SRC interactions, and plays a key regulatory role in rice seed germination and seedling growth [62]. In our result, both were up-regulated in transcription level in the knockout lines compared to WT. Meanwhile, it has been shown that *OsTPP7* is involved in T6P metabolism and catalyzes the conversion of T6P to trehalose, thereby allowing increased starch mobilization in the form of easily fermentable sugar, which ultimately enhances coleoptile elongation and embryo germination [2]. Similarly, in our research, better developed coleoptiles and higher levels of *OsTPP7* expression were observed in the knockout lines (Appendix A). This suggests, therefore, that knockout of *NAL11* could improve submergence tolerance in rice by affecting the transcript levels of these stress-responsive genes.

The phytohormone auxin has long been known to be important in stimulating coleoptile elongation and rapid seedling growth in the air [31], but little is known about its role in influencing the rice coleoptile elongation under water. To understand the role of plant hormones in rice under air and hypoxic conditions, we carried out some analyses on ABA, GA and auxin and related genes under submerged conditions. ABA is a well-documented stress hormone that accumulates in response to stress [21]. Consequently, ABA levels in the *NAL11* knockout lines gradually increased after 48 h of submergence treatment. We also found that ABA and GA levels were reversed at 48 h and 72 h after submergence treatment regardless of in both knockout lines and WT (Figure 6). It has previously been shown that the vivipary phenotype in maize kernels due to ABA deficiency can be reversed through inhibition of GA synthesis, demonstrating the role of GA in antagonizing the action of ABA. In this study, this phenomenon was well explained by the expression of genes related to the biosynthesis and metabolism of ABA and GA (Appendix A). These results suggest that *NAL11* negatively regulates the antagonistic effects of ABA and GA by mediating the activities of a number of enzymes involved in ABA- and GA-related biosynthesis. Thus, we can reasonably infer that the antagonistic regulation of GA and ABA metabolism mainly occurs by activating and repressing the opposing metabolic genes (*NCED/GA2ox or ABA8ox/GA3ox* family) to maintain a hormonal balance during plant growth and development and to respond to environmental cues. Auxin is well known for promoting coleoptile elongation and rapid seedling growth during germination [62], but little is known about its role in rice germination and seedling establishment under submergence. A recent report has also demonstrated that auxin is required for rice seed germination under submergence. The results indicate that auxin availability and transport play a critical role in determining the final coleoptile length in Japonica rice [32]. In submerged seeds, the knockout lines had higher levels of endogenous auxin than WT, which is consistent with the fact that the expression of four auxin biosynthesis genes, *YUCCA2*, *YUCCA3*, *YUCCA6* and *TAA1*, was significantly increased in seedlings of hypoxic knockout lines in comparison to WT (Appendix A), consistent with the observed phenotype (Appendix A) and the previous studies. This suggests that *NAL11* may regulate auxin biosynthesis, enhancing coleoptile elongation under submergence. Whether or not the auxin transport or distribution in hypoxic rice seedlings would be influenced by excessive accumulation of endogenous free IAA and affect the submergence tolerance remains to be investigated.

Finally, ABA plays a dual role in growth regulation, promoting growth at low concentrations and inhibiting it at high concentrations [25,63]. In our study, *NAL11* knockout lines exhibited enhanced ABA sensitivity (Figure 7A,B). This phenomenon was further confirmed by experiments using low concentrations (0.001 µM) of exogenous ABA treatment, which stimulated coleoptile growth. However, when exposed to higher concentrations of exogenous ABA, coleoptile growth was inhibited. When the optimal concentration for plant growth was exceeded, the addition of exogenous ABA led to the inhibition of coleoptile growth in both the knockout lines and the WT when treated with 0.1 µM and 1.0 µM ABA (Figure 7A), in agreement with which is consistent with the previous studies [64]. The increased sensitivity of the knockout lines to exogenous ABA (Figure 7A,B) implies an enhanced responsiveness of these lines to ABA. This enhanced sensitivity was supported by increased expression of ABA signaling genes, including *OsPYL* [1] and *OsbZIP72* [65,66] (Appendix A). These results indicate that *NAL11* may balance ABA signaling and GA biosynthesis to modulate coleoptile elongation and stress responses under submergence.

## 4. Materials and Methods

### 4.1. Bioinformatics Analysis of NAL11

The information and sequences of NAL11 homologues were retrieved from the NCBI database (http://www.ncbi.nlm.nih.gov), accessed on 3 November 2023. Utilizing ClustalW with default parameters, multiple sequence alignments were performed on protein sequences, followed by manual adjustments. Subsequently, a phylogenetic tree was constructed with aligned protein sequences using MEGA (v11.0) software, employing the neighbor-joining (NJ) method. Bootstrap values, derived from 1000 iterations, were calculated to assess the robustness of the tree [67]. For the identification of conserved motifs within protein sequences, online MEME (v5.5.7) (http://meme-suite.org/) was utilized with default parameters. Putative *cis*-acting elements were identified by analyzing the 2000-bp promoter region sequences of these paralogous genes, obtained from the NCBI database, using PlantCARE (http://bioinformatics.psb.ugent.be/webtools/plantcare/html/). TBtools software (v1.120) facilitated the visualization of the phylogenetic tree, conserved motifs, gene structure, domains, and *cis*-acting elements in promoters [68].

### 4.2. Plant Materials and Growth Conditions

Zhonghua 11 (ZH11) is a japonica rice (*Oryza sativa* L.) variety used as the WT plant and the recipient for genetic transformation in this study. The T_1_ and T_2_ generation knockout lines were consecutively assayed for the target gene and the hygromycin resistance gene. Two stable T_3_-generation knockout lines *(ko-nal11-1* and *ko-nal11-2*) without hygromycin were selected for follow-up studies. All plants were cultivated under natural conditions in the experimental field at South China Agricultural University (Guangzhou, Guangdong, China, 23.13° N, 113.27° E). The WT and knockout lines without hygromycin were planted in a randomized block design. Each plot consisted of six rows with six plants per row at a planting interval of 20 cm × 20 cm. Field management was in accordance with normal agricultural practices.

### 4.3. Construction of Transgenic Plants

To achieve the knockout lines of *NAL11*, target sites were designed through the online tool Clustered Regularly Interspaced Short Palindromic Repeats (CRISPR)-GE/targetDesign (http://skl.scau.edu.cn/) [69]. Genome-targeting constructs were then prepared using the pRGEB32 vector [70]. The structure of the *NAL11* CRISPR/Cas9 knockout vector is shown in Appendix A. Pre-cultured Zhonghua 11 (ZH11) seeds were immersed in the Agrobacterium suspension by gently inverting the tube for 1.5 min, then blotted dry with a sterilized filter paper to remove excess bacteria. These seeds were transferred onto a sterilized filter paper (9-cm diameter) that had been moistened with 0.5 mL of AAM medium placed on 2N6-AS medium solidified with 0.4% Gelrite. After 3 days of co-cultivation at 25 °C in the dark, seeds were washed five times in sterile water and then washed once in sterile water containing 500 mg L^−1^ carbenicillin (Wako Pure Chemicals, Osaka, Japan) to remove Agrobacterium. The seeds were rapidly blotted dry on a sterilized filter paper and cultured on N6D medium containing 50 mg L^−1^ hygromycin and 400 mg L^−1^ carbenicillin under continuous light at 32 °C for 2 weeks. Proliferating calli emerging from the scutellum were transferred to RE-III medium. Plantlets emerging from the calli were transferred to HF medium to induce roots. For validation, the target sites of T_1_ and T_2_ generation plants were sequenced and analyzed using CRISPR-GE/DSDecodeM [71].

To obtain the expression profile of *NAL11*, the 2000 bp promoter sequence was identified from Ensemble Plants (*LOC_Os07g09450*) and amplified from rice genomic DNA. The *pCAMBIA1305* vector was double digested with *SacI* and *BglII* endonucleases, followed by recombination to generate the *pNAL11::GUS* vector. The *pNAL11::GUS* vector was introduced into Agrobacterium tumefaciens strain EHA105, and the transformation procedure described above was used to generate transgenic rice lines. Transgenic plants carrying the *pNAL11::GUS* construct were screened and confirmed by PCR. All primers used in the construction process are detailed in Appendix A.

### 4.4. Evaluation of Germination Rate and Coleoptile Length

Seeds of each line were grown in the field, and their seeds were harvested 45 days after heading, air dried, and stored at 42 °C for 7 days to break dormancy. Three independent biological replicates of 30 seeds per replicate were then sterilized with 1.5% (*v*/*v*) sodium hypochlorite and subsequently incubated in a 9-cm diameter Petri dish. Seeds were considered germinated when the white embryo protrusion was visible, at which point the germination percentage was then calculated. For the anoxic experiments, five seeds were placed in a test tube (diameter: 2.7 cm; height: 11.7 cm) filled with distilled water to simulate anaerobic conditions [72]. All germination experiments were performed in a controlled environment at 28 °C under a 12 h/12 h light/dark cycle. After 4 days, a WinRHIZO root image analysis system (Regent Instruments Inc., Québec, QC, Canada) was used to measure the coleoptile length (CL), coleoptile surface area (CSA), and coleoptile diameter (CD). Three biological replicates were tested.

### 4.5. Germination Test by Exogenous Application of ABA

A total of 15 seeds of the WT and knockout line (*nal11*) were anaerobically incubated at 28 °C with a gradient concentration of ABA solution as treatment and with pure water as control, respectively. The concentration of the ABA solution was adjusted to 0.001, 0.01, 0.1, 0, 1, and 10 µM, respectively. Photographs were taken and coleoptile lengths were measured after 4 d of incubation.

### 4.6. Measurement of Endogenous ABA, GA and IAA Levels

To measure the endogenous levels of ABA, GA and IAA, the seeds were prepared at 24, 48 and 72 HAI (hours after imbibition) under anaerobic stress. Liquid nitrogen-frozen germinated seeds (50 mg fresh weight) were ground to powder and extracted with the traction method (methanol/water/formic acid = 15:4:1, *V/V/V*). The extracts were vortexed and centrifuged at 4694× *g* at 4 °C for 10 min. The supernatants were dried by evaporation under the flow of nitrogen gas at room temperature, then dissolved in 200 μL of methanol. The sample extracts were analyzed using an LC-ESI-MS/MS system (HPLC, Shim-pack UFLC SHIMADZU CBM30A system; MS, Applied Biosystems 6500 Triple), and the data were analyzed by Zoonbio Biotechnology Co., Ltd., Nanjing, China. Three replicates of each assay were performed [73].

### 4.7. Analysis of Physiological Parameters Related to Submergence Stress

Ten seeds each from the WT and knockout lines were anaerobically incubated at 28 °C, respectively. The treatment time was set at 0 h, 12 h, 24 h, 48 h, 72 h and 96 h, respectively, and the analysis of catalase (CAT), superoxide dismutase (SOD), peroxidase (POD) activity and content of malondialdehyde (MDA) and H_2_O_2_ content was performed as previous described [74] and adjusted. For all assays, the data for each time point represent the average of at least three biological replicates. Calculations were carried out according to the equations recommended by the manufacturer (Nanjing Jiancheng Bioengineering Institute, Nanjing, China).

### 4.8. RNA Extraction and Analysis of Gene Expression

Total RNA from germinated seeds was extracted using TRIzol reagent (R401-01, Vazyme, Nanjing, China). The first-strand cDNA was synthesized from 600 ng total RNA using a reverse transcription kit (R133-01, Vazyme). The qRT-PCR reaction was performed using ChamQ Universal SYBR qPCR Master Mix (Q711-03, Vazyme) on an ABI Step One Plus Real-Time PCR System (Applied Biosystems, Foster City, CA, USA) [75]. Normalized transcript levels were calculated using the comparative 2^−ΔΔCT^ method [76], with the *OsActin* serving as an internal control. Five biological replicates were performed. All primers used for qRT-PCR are listed in Appendix A.

The developing seeds of *pNAL11::GUS* transgenic plants were collected and detected according to the previous method [77]. The developing seeds of *pNAL11::GUS* transgenic plants were collected and incubated in GUS staining buffer (750 μg·mL-1 X-gluc, 10 mM EDTA, 3 mM K3Fe(CN)6, 100 mM NaPO_4_ pH 7, and 0.1% Nonidet-P40) at 37 °C for 6 h. The samples were then transferred to 70% ethanol to remove chlorophyll. Finally, photographs were taken using a ZEISS stereomicroscope.

### 4.9. Statistical Analysis

Statistical analysis of the data was performed using the Prism 8.3.0 software package. Student’s *t*-tests were employed, with statistical significance set at *p* < 0.05. Significant differences between means are indicated by asterisks (* *p* < 0.05, ** *p* < 0.01, *** *p* < 0.001). All data are presented as mean ± standard deviation (SD), with “n” representing the sample size.

## 5. Conclusions

Bioinformatic analysis reveals that the *NAL11* gene belongs to the family of HSPs containing the DnaJ structural domain. Knockout of *NAL11*, which regulates the expression of genes involved in ABA, GA and auxin biosynthesis, catabolism and signaling pathways, increases the activities of many antioxidant defence enzymes to maintain ROS balance and improve tolerance to submergence stress in rice. Additionally, knockout of *NAL11* significantly increased sugar metabolism and the expression of expansin genes in rice seeds, which promoted the elongation of rice coleoptiles. Taken together, these molecular and physiological changes resulted in improved tolerance to submergence stress in rice. These findings not only deepen our understanding of the function of *NAL11*, but also provide a solid foundation for future research aimed at improving crop tolerance to flooding, potentially leading to more resilient agricultural practices in flood-prone areas.

## Figures and Tables

**Figure 1 plants-13-02593-f001:**
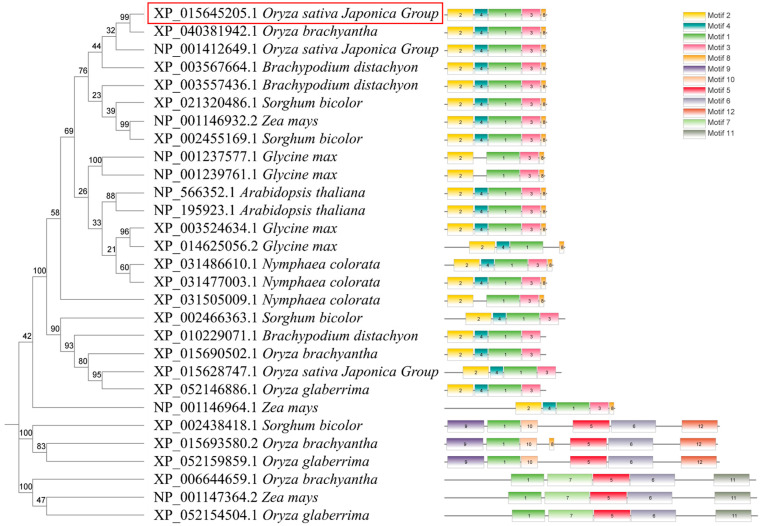
Evolutionary tree of NAL11 homologs with Oryza sativa Japonica, Brachypodium distachyon, Sorghun bicolor, Zea mays, Glycine max, Nymphaea colorata, Arabidopsis, and wild rice (Oryza brachyantha, Oryza glaberrima). XP_015645205.1 is the accession number of NAL11 in NCBI, which is highlighted in the red box.

**Figure 2 plants-13-02593-f002:**
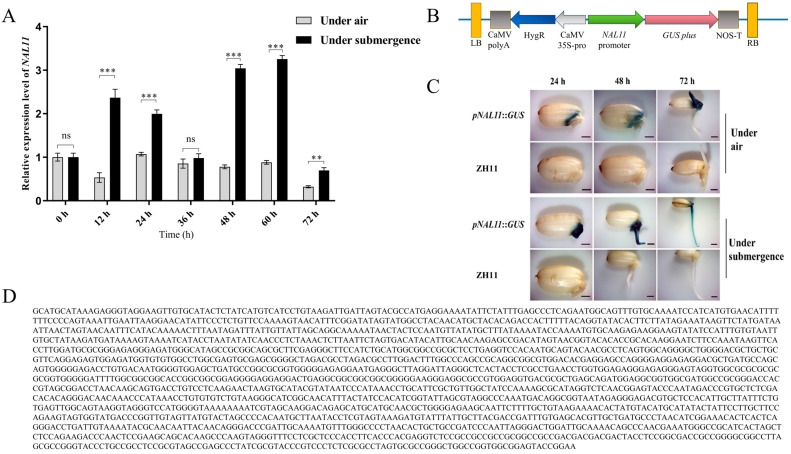
Spatiotemporal expression analysis of *NAL11*. (**A**) Transcription levels of *NAL11* in germinating seeds of ZH11 under aerobic and submerged conditions using quantitative reverse transcription polymerase chain reaction (RT-PCR). Gene expression was normalized to that of *OsActin*, with relative expression levels represented as fold change relative to the expression level of *NAL11* at 0 h. (**B**) Schematic structure of the expression vector for pro*_NAL11_*:GUS. (**C**) β-glucuronidase (GUS) staining of seeds after 24 h, 48 h, and 72 h of submergence stress. Scale Bar, 0.5 cm. (Data are presented as mean ± SD, *n* = 5; significant differences were determined by two-tailed Student’s *t*-tests. ** *p* < 0.01, *** *p* < 0.001, ns, no significance). (**D**) 2000 bp promoter sequence of *NAL11*.

**Figure 3 plants-13-02593-f003:**
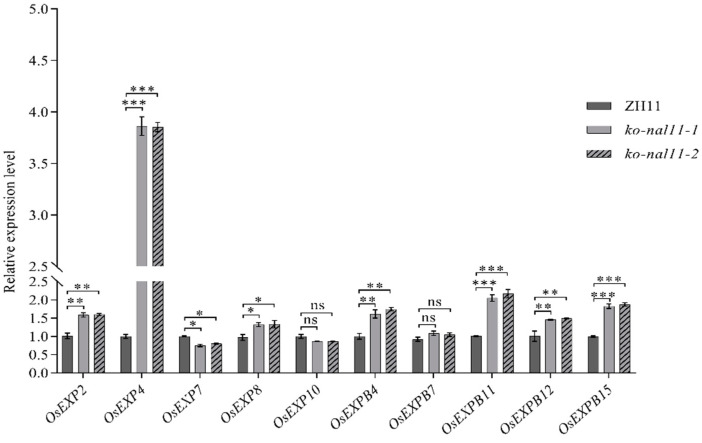
Relative expression level of expansin genes in ZH11 and knockout lines at 48 h after submergence. (Data are presented as mean ± SD, *n* = 5; significant differences were determined by two-tailed Student’s *t*-tests. * *p* < 0.05, ** *p* < 0.01, *** *p* < 0.001, ns, no significance).

**Figure 4 plants-13-02593-f004:**
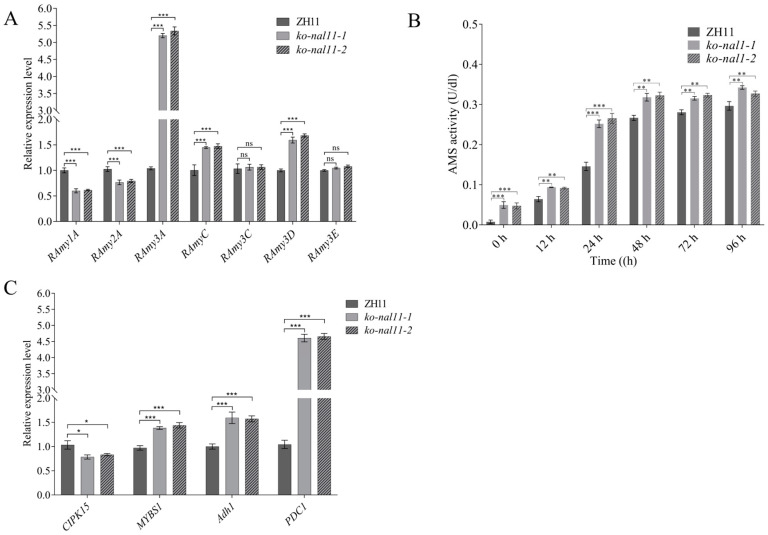
*NAL11* is involved in the sugar and energy pathway. (**A**) Quantitative RT-PCR analysis of nine α-amylase family genes in seeds of ZH11 and knockout lines at 48 h after submergence treatment. (**B**) α-AMS activity at different stages. (**C**) RT-qPCR analysis of sugar and energy metabolism genes at 48 h after submergence treatment, respectively. (Data are presented as mean ± SD, *n* = 5; significant differences were determined by two-tailed Student’s *t*-ests. * *p* < 0.05, ** *p* < 0.01, *** *p* < 0.001, ns, no significance).

**Figure 5 plants-13-02593-f005:**
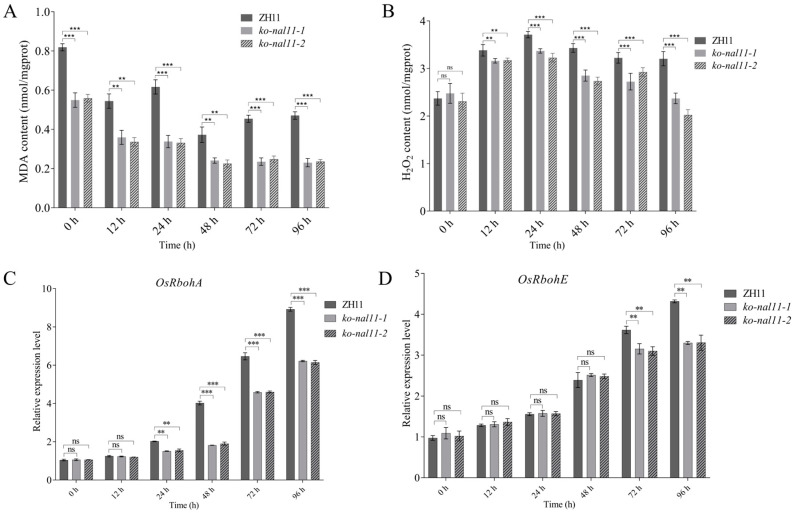
Analysis of MDA and H_2_O_2_ content, transcript levels of ROS-production gene, *OsRbohA* and *OsRbohE* of WT and transgenic plants under normal and submerged conditions. (**A**) MDA content. (**B**) H_2_O_2_ content. (**C**) *OsRbohA*. (**D**) *OsRbohE*. (Data are presented as mean ± SD, *n* = 5; significant differences were determined by two-tailed Student’s *t*-tests. ** *p* < 0.01, *** *p* < 0.001, ns, no significance).

**Figure 6 plants-13-02593-f006:**
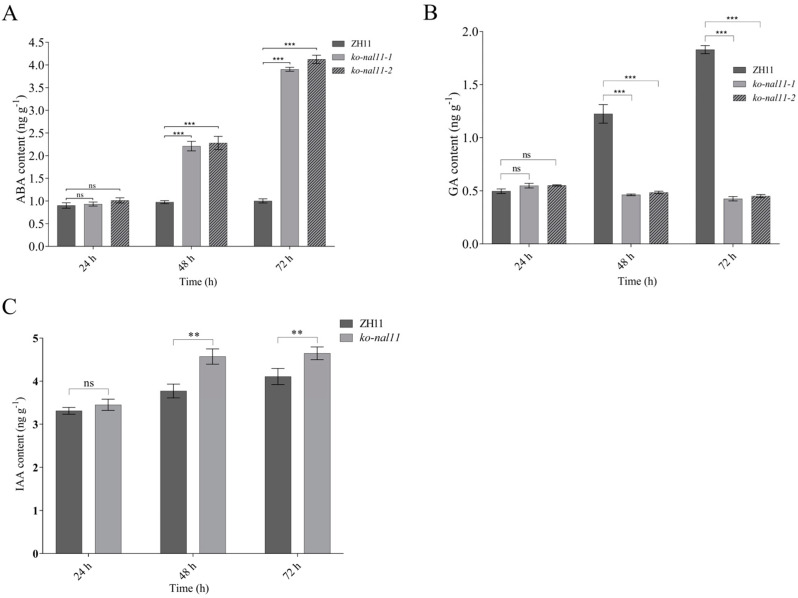
Knockout of *NAL11* affects the levels of ABA, GA and IAA. (**A**–**C**) Content of endogenous ABA, GA and IAA at 24 h, 48 h, and 72 h after submergence. (Data are presented as mean ± SD, *n* = 5; significant differences were determined by two-tailed Student’s *t*-tests. ** *p* < 0.01, *** *p* < 0.001, ns, no significance).

**Figure 7 plants-13-02593-f007:**
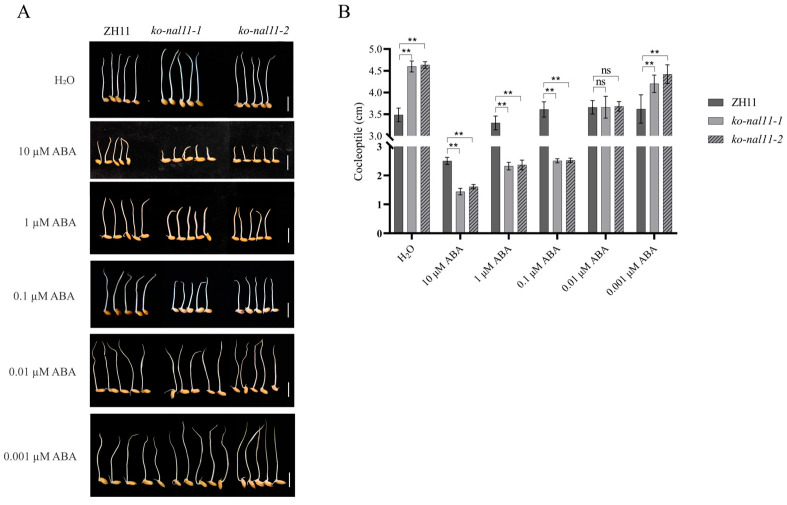
Knockout of *NAL11* shows sensitivity to ABA. (**A**) Representative images of coleoptile length in response to different concentrations of ABA (0 µM, 0.001 µM, 0.01 µM, 0.1 µM, 1 µM, and 10 µM, respectively) after 4 d of submergence for both WT and knockout lines. Scale bars: 1 cm. (**B**) Comparison of coleoptile lengths in WT and knockout lines in response to control (H_2_O) and ABA treatments after 4 d of submergence. (Data are presented as mean ± SD, *n* = 5 biologically independent samples; significant differences were determined by two-tailed Student’s *t*-tests. ** *p* < 0.01, ns, no significance).

## Data Availability

All of the datasets are included within the article and its additional files.

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
