# Peer review of "Enhancing Coleoptile Length of Rice Seeds under Submergence through NAL11 Knockout"

_plants, 2024, doi:10.3390/plants13182593_

Round 1

Reviewer 1 Report (Previous Reviewer 3)

Comments and Suggestions for Authors

The proposed changes have been made.
Congratulations

Author Response

Reviewer #1:

The proposed changes have been made.
Congratulations.

Reply:

Thank you very much for your positive feedback and for acknowledging the changes we have made. We are delighted that the revisions meet your expectations. Your insightful comments have significantly improved the quality of our manuscript, and we greatly appreciate the time and effort you have taken to review our work.

Reviewer 2 Report (New Reviewer)

Comments and Suggestions for Authors

Thank you for your manuscript entitled "Enhancing coleoptile length of rice seeds under submergence through NAL11 knockout " which you submitted to Plants.

I think this manuscript shows useful information and appropriate for “Plants” publication.

However, it is necessary for revision according to journal instruction. 

Author Response

评论者 #2

感谢您向 Plants 提交的题为“通过 NAL11 敲除增强淹没下水稻种子的胚芽鞘长度”的手稿。

我认为这份手稿显示了有用的信息,适合“植物”的出版。

但是,有必要根据期刊指示进行修订。

请参阅附件。

答:

感谢您对我们题为“通过 NAL11 敲除增强水稻种子在淹没下的胚芽鞘长度”的手稿的积极反馈。感谢您的宝贵意见和提供的指导。

我们很高兴地通知您,我们已经仔细审查了期刊说明和附件中的建议。所有必要的修订都已相应地进行。

再次感谢您的支持,我们期待着您的反馈。

Reviewer 3 Report (New Reviewer)

Comments and Suggestions for Authors

The work “Enhancing coleoptile length of rice seeds under submergence through NAL11 knockout” represents a kind of continuation in the description for the role of this gene in rice development and stress (abiotic, in this case submergence/flooding). In general, the work is quite well presented and supported throughout all its sections and could be of relevance in the area. However, several points need to be clarified.

Introduction.

The section is well developed, descriptive, clear, and covers interesting aspects associated to abiotic stress in rice, including plant´s response mechanisms. In the last paragraph, authors describe their previous work with this gene, in which linkage to cell cycle and development were described; results included involvement in GA biosynthesis and stomata formation/function. Whereas NAL11´s role, as indicated in ref#36, was derived from an evident phenotype associated to a mutation (923-1552 deletion), the current aim is, however, associated to the role of this gene in stress responses; so, why authors decided going further into this direction? In other terms, whereas the section is well conducted and built, considering that NAL11 was previously studied (by this same authors), how did they get the point to make a “detour” into this (submergence/flooding) area? May be this could be the only weak point in the section that I suggest improving.

Results.

2.1. The section begins with a phylogenetic study for NAL11 including 28 homologs. In the current format, this sounds some contradictory (to me) if authors already had made a deep study on this gene (ref#36), which in addition is not the first description for the same gene. Why these comparisons are needed and included in the present work? Which is the idea getting these analyses? This is the first paragraph in the section, and it will acquire more solidity if a clear starting point is announced here, ideally establishing differences with previous works. This time authors included response elements in promoters to link NAL11 to stress?

2.2. NAL11::GUS construct. Authors indicate that a “promoter sequence of approximately 2,000 bp was identified from Ensemble Plants (LOC_Os07g09450) and amplified from rice genomic DNA” to generate the chimeric construct driving the GUS reporter gene. Approximately 2 Kbp is an ambiguous term to describe the construct. Please, include a map and the sequences involved (a supplementary figure is encouraged). In Fig 2A, mRNA levels corresponded to determinations using RNA extractions from whole explants regardless the developmental stage (important differences in coleoptiles´ length are seen in 2B)? In Fig 2B, what is CK? Caption in Figure 2 is not clear.

2.3 – 2.7. Starting in section 2.3, two knock-out ZH11 lines are the focus of the next assays and characterizations regarding plant´s (anaerobic) stress responses. I couldn´t find any description of the (CRISPR-Cas) mutagenized lines (nal11-1 and nal11-2) in the main or in supplementary files. Am I right? If so, a complete description of the NAL11 edited lines is needed, including NAL11 mRNA levels for both lines. As for the previous point (2.2), maps for the constructs should be desirable (did you use 1 sgRNA, or 2 gRNAs?; gRNAs´ sequences? What kind of mutation was induced?). Please, take care on these descriptions because I found similar elements in ref#36, including the use of ZH. Currently, it seems confusing.

Discussion.

As for the general of the work, the section seems well constructed; however, all this structure is weak if authors do not generate a clear picture of the experimental procedures, including the generated materials used for these experiments.

Anyway, the interpretation of results should be landed into a position in which the responses under evaluation are correlated to the effect of mutations in NAL11, but this is not necessarily the meaning for this gene as an integrator those pathways. For instance, as indicated in previous works for the gene, it will be associated, in some way and/or extent, to several stress response mechanisms, but this is not meaning that is the key element for those responses.

Author Response

Reviewer #3:

The work “Enhancing coleoptile length of rice seeds under submergence through NAL11 knockout” represents a kind of continuation in the description for the role of this gene in rice development and stress (abiotic, in this case submergence/flooding). In general, the work is quite well presented and supported throughout all its sections and could be of relevance in the area. However, several points need to be clarified.

Reply:

We thank this reviewer for the comments and insights, especially for confirming the importance of this study. We have revised the manuscript according to your suggestions and hope that the revised manuscript will meet the journal’s requirements. The detailed responses/corrections are listed below, point by point.

Introduction.

The section is well developed, descriptive, clear, and covers interesting aspects associated to abiotic stress in rice, including plant´s response mechanisms. In the last paragraph, authors describe their previous work with this gene, in which linkage to cell cycle and development were described; results included involvement in GA biosynthesis and stomata formation/function. Whereas NAL11´s role, as indicated in ref#36, was derived from an evident phenotype associated to a mutation (923-1552 deletion), the current aim is, however, associated to the role of this gene in stress responses; so, why authors decided going further into this direction? In other terms, whereas the section is well conducted and built, considering that NAL11 was previously studied (by this same authors), how did they get the point to make a “detour” into this (submergence/flooding) area? May be this could be the only weak point in the section that I suggest improving.

Reply:

Thank you again for your valuable comments and suggestions. We appreciate the opportunity to clarify our rationale for extending the investigation of NAL11 to submergence and stress responses.

While our previous work (ref#36) focused on the role of NAL11 in plant architecture and development, particularly its involvement in gibberellin biosynthesis, our current study focuses on into its potential role in abiotic stress responses, specifically submergence and flooding. The decision to investigate the role of NAL11 in stress tolerance arose from two key observations:

Emerging Evidence for HSPs in Stress Responses: Heat shock proteins (HSPs), such as those in the DnaJ family to which NAL11 belongs, have been shown to play important roles in several abiotic stress responses, including oxidative stress, high temperature and anoxia. Although the role of NAL11 in growth and development had been established, we hypothesised that its function might extend beyond this, especially given its regulatory role in key processes related to metabolism and hormone regulation, which are critical in stress tolerance mechanisms. It was therefore logical for us to investigate the possibility that NAL11 contributes to submergence tolerance in rice, particularly under hypoxic conditions, where heat shock proteins have been implicated in other species.

The Link Between the Role of NAL11 in Plant Architecture and Stress Responses:

Coleoptile and mesocotyl elongation are critical for rice survival under submerged conditions, and these processes are intricately linked to hormonal regulation (ABA, GA, auxin). Our previous studies have showed that NAL11, a gene encoding a heat shock protein containing the DnaJ structural domain, regulates rice plant architecture and is involved in GA metabolism, demonstrating the influence of NAL11 on hormone regulation. This prompted us to investigate whether the regulatory function of NAL11 in growth would intersect with stress response pathways, especially under environmental stresses such as flooding, which can challenge plant survival.

We appreciate your suggestion to clarify this transition in the manuscript. We have revised the last paragraph of the introduction to better explain why we extended our research on NAL11 from growth regulation to stress responses, making the connection between our previous findings and the current focus more explicit.

Thank you again for your constructive feedback.

Results.

2.1 The section begins with a phylogenetic study for NAL11 including 28 homologs. In the current format, this sounds some contradictory (to me) if authors already had made a deep study on this gene (ref#36), which in addition is not the first description for the same gene. Why these comparisons are needed and included in the present work? Which is the idea getting these analyses? This is the first paragraph in the section, and it will acquire more solidity if a clear starting point is announced here, ideally establishing differences with previous works. This time authors included response elements in promoters to link NAL11 to stress?

Reply:

Thank you for your insightful feedback regarding the phylogenetic study in our manuscript. We appreciate your concerns and would like to clarify the rationale for including the homolog comparison and the new analysis.

Although we have previously studied NAL11 (ref#36), the current phylogenetic analysis includes new species, specifically aquatic plants and additional Poaceae species, which have not been examined in previous work. This broader phylogenetic context is crucial for understanding whether NAL11 and its homologs have adapted specific functions related to stress responses, especially in aquatic and monocot species, which may show unique evolutionary adaptations to environmental stresses such as hypoxia and submergence. The new phylogenetic analysis aims to explore potential functional diversification among NAL11 homologs in various species, including those that are adapted to aquatic environments. This comparison will help us to understand whether the mechanisms observed in rice are shared or distinct across different species.

As you noted, this time we included an analysis of response elements in the promoter region to provide additional evidence for the involvement of NAL11 in stress responses. This analysis complements the phylogenetic study by suggesting potential regulatory mechanisms by which NAL11 and its homologs might respond to environmental stresses, supporting our hypothesis that NAL11 plays a role in stress adaptation.

Thank you again for your thoughtful comments, which have helped us refine our manuscript.

2.2 NAL11::GUS construct. Authors indicate that a “promoter sequence of approximately 2,000 bp was identified from Ensemble Plants (LOC_Os07g09450) and amplified from rice genomic DNA” to generate the chimeric construct driving the GUS reporter gene. Approximately 2 Kbp is an ambiguous term to describe the construct. Please, include a map and the sequences involved (a supplementary figure is encouraged). In Fig 2A, mRNA levels corresponded to determinations using RNA extractions from whole explants regardless the developmental stage (important differences in coleoptiles´ length are seen in 2B)? In Fig 2B, what is CK? Caption in Figure 2 is not clear.

Reply:

Thank you for your valuable comments and suggestions.

Regarding the pNAL11::GUS construct:

We have addressed your concern about the use of the term "approximately 2K bp" by including a detailed map of the construct and the exact sequences involved in the Fig. 2B, D. This map provides a clear representation of the promoter region and the chimeric construct driving the GUS reporter gene. The exact length of the promoter sequence has also been provided to avoid ambiguity.

Regarding the mRNA levels in Figure 2A and the impact of developmental stage:

In Figure 2A, we highlight the difference in expression levels of the NAL11 gene under hypoxic (submerged) and non-hypoxic (air) conditions. The experimental results clearly showed that the expression level of NAL11 was significantly increased under hypoxic conditions, especially after 48 h and 60 h treatments, which was closely related to the rapid elongation stage of the embryonic sheath. This phenomenon suggests that NAL11 may play a key role in regulating the elongation and growth of the embryo sheath under hypoxic stress. This point fully justifies the central point we want to express, namely the importance of NAL11 in hypoxic germ sheath elongation.

Whether there is a consistent pattern of change in mRNA levels at different developmental stages, although our current experimental data are insufficient to draw definitive conclusions, this is not the main goal of this study. We have mainly focused on the expression differences between hypoxic and non-hypoxic conditions, which is indeed reflected in our experimental results.

We believe that the current experimental results are sufficient to support our speculation about the function of the NAL11 gene in hypoxic environments and that the results are reliable and plausible. If necessary, we will conduct more detailed studies in the future, but this is beyond the scope of this study.

Regarding the CK (Control) designation:

We have clarified in the figure legend that CK refers to the wild-type ZH11 as the control in our experiments to ensure that the terminology is clear throughout the manuscript.

Once again, we appreciate your feedback and the opportunity to clarify our findings. We hope that this response satisfactorily addresses your concerns.

2.3-2.7. Starting in section 2.3, two knock-out ZH11 lines are the focus of the next assays and characterizations regarding plant´s (anaerobic) stress responses. I couldn´t find any description of the (CRISPR-Cas) mutagenized lines (nal11-1 and nal11-2) in the main or in supplementary files. Am I right? If so, a complete description of the NAL11 edited lines is needed, including NAL11 mRNA levels for both lines. As for the previous point (2.2), maps for the constructs should be desirable (did you use 1 sgRNA, or 2 gRNAs?; gRNAs´ sequences? What kind of mutation was induced?). Please, take care on these descriptions because I found similar elements in ref#36, including the use of ZH. Currently, it seems confusing.

Reply:

Thank you for your thorough review and valuable suggestions. We have made the necessary revisions as per your comments.

Regarding the description of the CRISPR-Cas9 mutagenized lines (nal11-1 and nal11-2):

You are correct in noting that a detailed description of the CRISPR-Cas9 edited lines was previously missing. We have now included a comprehensive description of the NAL11 knockout lines in the main text and supplementary materials. This includes details of the mutagenesis method, the sequences of the gRNAs used (one gRNA), and the types of mutations induced. Additionally, we have provided the NAL11 mRNA levels for both the nal11-1 and nal11-2 lines to further characterize the knock-out phenotypes. The changed part is marked in red in the revised manuscript.

About the constructs:

We have included maps of the CRISPR-Cas9 constructs used to generate the knockout lines, as well as the gRNA sequences, in the supplementary files (Supplementary Figure 3 and Table 1). This should clarify any confusion and provide a full understanding of the experimental setup. The specific mutations introduced in each line are also detailed.

We sincerely appreciate your comments, which have helped to improve the clarity and completeness of our manuscript.

Discussion.

As for the general of the work, the section seems well constructed; however, all this structure is weak if authors do not generate a clear picture of the experimental procedures, including the generated materials used for these experiments.

Anyway, the interpretation of results should be landed into a position in which the responses under evaluation are correlated to the effect of mutations in NAL11, but this is not necessarily the meaning for this gene as an integrator those pathways. For instance, as indicated in previous works for the gene, it will be associated, in some way and/or extent, to several stress response mechanisms, but this is not meaning that is the key element for those responses.

Reply:

Thank you for your valuable comments on our articles. We take your feedback very seriously and have carefully considered your suggestions for incremental improvements. We would like to respond to each of your comments below:

Regarding the clarity of experimental procedures and materials: We fully understand your concerns regarding the description of experimental procedures and materials. During the revision process, we have made detailed additions to the experimental section, especially regarding the description of the generated materials (e.g., mutants). We hope that these improvements will make our experimental design clearer, so that readers can better understand the experimental methods and procedures we used.

Regarding the interpretation of the results in relation to the role of the NAL11 gene: Thank you for reminding us to be more cautious in our interpretation of the NAL11 gene. We have revised the Discussion section to further elaborate on the possible role of the NAL11 gene in different stress response mechanisms. We acknowledge that the NAL11 gene may play a role in these pathways, but is not the only key factor. In the Discussion, we have reviewed the interpretation of the data to ensure that we avoid over-interpretation and more accurately discuss the relevance of NAL11 to the stress response.

Thank you again for your constructive comments, your feedback has been instrumental in improving our paper. We look forward to your further review and sincerely hope that our revisions will meet your requirements.

Round 2

Reviewer 3 Report (New Reviewer)

Comments and Suggestions for Authors

I have reviewed the R1 document and observed that all the comments I made on the original version have been incorporated or addressed by the authors. Thank you for the current presentation, which certainly merits publication in Plants.

I have no further comments to add to the present version.

This manuscript is a resubmission of an earlier submission. The following is a list of the peer review reports and author responses from that submission.

Round 1

Reviewer 1 Report

Comments and Suggestions for Authors

In this manuscript, authors describe enhancing coleoptile length of rice seeds under submerg- ence through NAL11 knockout. This study shows well-planned exeperimental results, and these results are thought to be of interest to readers.

Some points to be considered:

1.      The introduction should present probable prior studies and related concepts that are appropriate and valid to understand the background and purpose of this study. However, this introduction presents several studies related to improving the length of rice plant leaves rather than related to the NAL11 gene. Therefore, more explanations and evidence for the NAL11 gene and heat shock protein are required in the introduction.

2.      In the introduction, after the sentence “Several genetic factors affect rice shoot growth” the sequences of prior studies presented should be listed to match the sequence of studies presented in the results.

-          “Several genetic factors affect rice shoot growth.” (Expansin gene, Anaerobic metabolic pathways, including glycolysis and fermentation, ROS, ABA...)

3.      The size of each figure is too small and difficult to identify. In particular, in the case of Figure 1, the color of the motif or the size and thickness of the font within the figure must be adjusted to make it easy to identify.

4.      In the conclusion, it seems necessary to add the results and implications of each experiment to the last sentence to provide validity and relevance to future experiments.

5.      Replacing the order of Figures 2 and 3 (results 2.2 and 2.3) seems appropriate to explain the role of NAL11 in elongation during submergence stress.

6.      It seems appropriate to change the order of Figures 4 and 5 (results 2.4 and 2.5) or to combine the two figures into one figure.

○ Please present the expression data of genes related to abiotic stress that occurs during immersion, and the results of the enzyme activity involved in the ROS antioxidant system generated at this time.

○ It must be explained that the ABA experiment was conducted on the basis that ROS and ABA were reported to interact [19,20].

7.      NAL11 knock out improves the plant's ability to respond to submergence stress and increases enzyme activity related to the ROS antioxidant system. Therefore, it seems necessary to increase the validity of the experiment by adding experimental results of the plant's stress response (ROS response experiment, H2O2 experiment).

8.      The results in Figure 6 (results 2.6) can fully explain the results in Figure 2 (results 2.2). Therefore, it is recommended that Figure 2 be explained as a supplementary figure.

9.      As a supplementary result to Figure 7, if we attach experimental results on the content of auxin and GA and the expression level of genes involved in each biosynthetic pathway, we believe that we can support the basis for the auxin motif and GA motif described in results 2.1.. Please consider.

10.  The discussion focuses on the research results and needs to compare them with previously reported results and prove the validity of the experimental results. However, since the concept that should have been explained in the introduction is described in the discussion, it is necessary to review and revise it again.

Reviewer 2 Report

Comments and Suggestions for Authors

In the manuscript entitled „Enhancing Coleoptile Length of Rice Seeds Under Submergence Through NAL11 Knockout” authors try to prove the role of  NAL11  in enhancing the tolerance of rice to submergence stress. It is very difficult to read this manuscript. Why this manuscript has a date 2022?

It is not known whether the authors analyse the amino acid or nucleotide sequence in chapter " Bioinformatics Characteristics ".

-            names of proteins should be written in capitals not italics, applies to the entire text

-            cis-regulatory elements not “cis-regulatory elements”

-            HSPs function in more than just the biological systems described by the authors

-            there are many incomprehensible sentences e.g. “Furthermore, under anoxic conditions, EXPA7 and EXPB12 are upregulated in rice coleoptiles, is required for rice shoot development”;  “Recent studies suggest an interaction between ROS and ABA interact and regulate each other”;

-            “To comprehensively characterize NAL11 and its homologous genes, we constructed a phylogenetic tree incorporating all 28 homologous proteins from diverse species.”  genes or proteins?

-            “Heat-shock proteins (HSPs) function as molecular chaperones and can either be newly synthesized or upregulated in response to stress in plants [26], including rice and Arabidopsis, particularly under anoxic conditions where HSP20 and HSP25.3-P are reduced [11].” This sentence is unclear, “reduced”?

-            Fig. 1 is illegible - this is the result of the analysis obtained from a given program, the description in the illustration does not contribute anything, analyses should be divided separately for DNA/gene and protein. Protein has no 5' and 3' ends.

-            “The amino acid sequence of the rice NAL11 gene” ???

-            There is no significant enhancement of POD activity in Nal11 knockout plants compared to WT, so it cannot be stated that “Moreover, NAL11 knockout

enhanced the activity of ROS scavenging enzymes, namely CAT, POD, and SOD in rice under submergence stress (Fig. 4B-D).

-            It is not clear to the reviewer why/on what basis the terms "tolerant genotype and intolerant genotype" appeared in the discussion.

-            The chapter “Construction of Transgenic Plants “ should be described in more detail

-            The method of introducing the construction into the callus should be described

-            GUS fusion constructs should be described in methods

Comments on the Quality of English Language

In the manuscript entitled „Enhancing Coleoptile Length of Rice Seeds Under Submergence Through NAL11 Knockout” authors try to prove the role of  NAL11  in enhancing the tolerance of rice to submergence stress. It is very difficult to read this manuscript. Why this manuscript has a date 2022?

It is not known whether the authors analyse the amino acid or nucleotide sequence in chapter " Bioinformatics Characteristics ".

-            names of proteins should be written in capitals not italics, applies to the entire text

-            cis-regulatory elements not “cis-regulatory elements”

-            HSPs function in more than just the biological systems described by the authors

-            there are many incomprehensible sentences e.g. “Furthermore, under anoxic conditions, EXPA7 and EXPB12 are upregulated in rice coleoptiles, is required for rice shoot development”;  “Recent studies suggest an interaction between ROS and ABA interact and regulate each other”;

-            “To comprehensively characterize NAL11 and its homologous genes, we constructed a phylogenetic tree incorporating all 28 homologous proteins from diverse species.”  genes or proteins?

-            “Heat-shock proteins (HSPs) function as molecular chaperones and can either be newly synthesized or upregulated in response to stress in plants [26], including rice and Arabidopsis, particularly under anoxic conditions where HSP20 and HSP25.3-P are reduced [11].” This sentence is unclear, “reduced”?

-            Fig. 1 is illegible - this is the result of the analysis obtained from a given program, the description in the illustration does not contribute anything, analyses should be divided separately for DNA/gene and protein. Protein has no 5' and 3' ends.

-            “The amino acid sequence of the rice NAL11 gene” ???

-            There is no significant enhancement of POD activity in Nal11 knockout plants compared to WT, so it cannot be stated that “Moreover, NAL11 knockout

enhanced the activity of ROS scavenging enzymes, namely CAT, POD, and SOD in rice under submergence stress (Fig. 4B-D).

-            It is not clear to the reviewer why/on what basis the terms "tolerant genotype and intolerant genotype" appeared in the discussion.

-            The chapter “Construction of Transgenic Plants “ should be described in more detail

-            Methods for preparing transgenic plants, both knockout analysis and promoter activity analysis, are very poorly described

-            The method of introducing the construction into the callus should be described

-            GUS fusion constructs should be described in methods

Reviewer 3 Report

Comments and Suggestions for Authors

General comments

The article is interesting, however there are some points that need to be improved, especially the materials and methods part.

Specific comments

Results

Figure 1: The size of the text in figure 1 is inappropriate. It is recommended that the figure be reconfigured to improve this aspect.

Objectives should be clearly defined. It is recommended that they be rewritten.

“Our previous study showed that the DnaJ domain-containing heat shock protein NAL11 regulates rice plant architecture and its interaction with gibberellin metabolism [28]. On this basis, this study aims to investigate the role of NAL11 in rice seedling emergence under flooded conditions. Our findings unveil a novel mechanism by which HSPs contribute to flood tolerance in rice, which lays a foundation for further investigations into flooding tolerance in rice.”

Materials and Methods:

The following sections should be better explained for a better understanding of the readers.

4.2. Plant Materials and Growth Conditions

4.4. Seed Germination Assay and ABA Treatment

4.5. Measurement of the Endogenous ABA Level

4.6. Analysis of physiological parameters related to submergence stress